# Locally Adaptive Bayesian Multivariate Time Series

**Daniele Durante**
Department of Statistical Sciences
University of Padua
Via Cesare Battisti 241, 35121, Padua, Italy
durante@stat.unipd.it

**Bruno Scarpa**
Department of Statistical Sciences
University of Padua
Via Cesare Battisti 241, 35121, Padua, Italy
scarpa@stat.unipd.it

**David B. Dunson**
Department of Statistical Science
Duke University
Durham, NC 27708-0251, USA
dunson@duke.edu

## Abstract

In modeling multivariate time series, it is important to allow time-varying smoothness in the mean and covariance process. In particular, there may be certain time intervals exhibiting rapid changes and others in which changes are slow. If such locally adaptive smoothness is not accounted for, one can obtain misleading inferences and predictions, with over-smoothing across erratic time intervals and under-smoothing across times exhibiting slow variation. This can lead to miscalibration of predictive intervals, which can be substantially too narrow or wide depending on the time. We propose a continuous multivariate stochastic process for time series having locally varying smoothness in both the mean and covariance matrix. This process is constructed utilizing latent dictionary functions in time, which are given nested Gaussian process priors and linearly related to the observed data through a sparse mapping. Using a differential equation representation, we bypass usual computational bottlenecks in obtaining MCMC and online algorithms for approximate Bayesian inference. The performance is assessed in simulations and illustrated in a financial application.

## 1  Introduction

### 1.1  Motivation and background

In analyzing multivariate time series data, collected in financial applications, monitoring of influenza outbreaks and other fields, it is often of key importance to accurately characterize dynamic changes over time in not only the mean of the different elements (e.g., assets, influenza levels at different locations) but also the covariance. It is typical in many domains to cycle irregularly between periods of rapid and slow change; most statistical models are insufficiently flexible to capture such locally varying smoothness in assuming a single bandwidth parameter. Inappropriately restricting the smoothness to be constant can have a major impact on the quality of inferences and predictions, with over-smoothing occurring during times of rapid change. This leads to an under-estimation of uncertainty during such volatile times and an inability to accurately predict risk of extremal events.

There is a rich literature on modeling a $p \times 1$ time-varying mean vector $\mu_t$, covering multivariate generalizations of autoregressive models (VAR, e.g. [1]), Kalman filtering [2], nonparametric mean regression via Gaussian processes (GP) [3], polynomial spline [4], smoothing spline [5] and Kernel smoothing methods [6]. Such approaches perform well for slowly-changing trajectories with

constant bandwidth parameters regulating implicitly or explicitly global smoothness; however, our interest is allowing smoothness to vary locally in continuous time. Possible extensions for local adaptivity include free knot splines (MARS) [7], which perform well in simulations but the different strategies proposed to select the number and the locations of knots (stepwise knot selection [7], Bayesian knot selection [8] or via MCMC methods [9]) prove to be computationally intractable for moderately large $p$. Other flexible approaches include wavelet shrinkage [10], local polynomial fitting via variable bandwidth [11] and linear combination of kernels with variable bandwidths [12].

Once $\mu_t$ has been estimated, the focus shifts to the $p \times p$ time-varying covariance matrix $\Sigma_t$. This is particular of interest in applications where volatilities and co-volatilities evolve through non constant paths. Multivariate generalizations of GARCH models (DVEC [13], BEKK [14], DCC-GARCH [15]), exponential smoothing (EWMA, e.g. [1]) and approaches based on dimensionality reduction through a latent factor formulation (PC-GARCH [16] and O-GARCH [17]-[18]) represent common approaches in multivariate stochastic volatility modeling. Although widely used in practice, such approaches suffer from tractability issues arising from richly parameterized formulations (DVEC and BEKK), and lack of flexibility resulting from the adoption of single time-constant bandwidth parameters (EWMA), time-constant factor loadings and uncorrelated latent factors (PC-GARCH, O-GARCH) as well as the use of the same parameters regulating the evolution of the time varying conditional correlations (DCC-GARCH). Such models fall far short of our goal of allowing $\Sigma_t$ to be fully flexible with the dependence between $\Sigma_t$ and $\Sigma_{t+\Delta}$ varying with not just the time-lag $\Delta$ but also with time. In addition, these models do not handle missing data easily and tend to require long series for accurate estimation [16]. Bayesian dynamic factor models for multivariate stochastic volatility [19] lead to apparently improved performance in portfolio allocation by allowing the dependence in the covariance matrices $\Sigma_t$ and $\Sigma_{t+\Delta}$ to vary as a function of both $t$ and $\Delta$. However, the result is an extremely richly parameterized and computationally challenging model, with selection of the number of factors via cross validation. Our aim is instead on developing continuous time stochastic processes for $\mu(t)$ and $\Sigma(t)$ with locally-varying smoothness.

Wilson and Ghahramani [20] join machine learning and econometrics efforts by proposing a model for both mean and covariance regression in multivariate time series, improving previous work of Bru [21] on Wishart Processes in terms of computational tractability and scalability, allowing more complex structure of dependence between $\Sigma(t)$ and $\Sigma(t + \Delta)$. Specifically, they propose a continuous time Generalised Wishart Process (GWP), which defines a collection of positive semi-definite random matrices $\Sigma(t)$ with Wishart marginals. Nonparametric mean regression for $\mu(t)$ is also considered via GP priors; however, the trajectories of means and covariances inherit the smooth behavior of the underlying Gaussian processes, limiting the flexibility of the approach in times exhibiting sharp changes.

Fox and Dunson [22] propose an alternative Bayesian covariance regression (BCR) model, which defines the covariance matrix of a vector of $p$ variables at time $t_i$, as a regularized quadratic function of time-varying loadings in a latent factor model, characterizing the latter as a sparse combination of a collection of unknown Gaussian process dictionary functions. More specifically given a set of $p \times 1$ vector of observations $y_i \sim N_p(\mu(t_i), \Sigma(t_i))$ where $i = 1, ..., T$ indexes time, they define

$$\text{cov}(y_i|t_i = t) = \Sigma(t) = \Theta\xi(t)\xi(t)^T\Theta^T + \Sigma_0, \quad t \in \mathcal{T} \subset \Re^+, \tag{1}$$

where $\Theta$ is a $p \times L$ matrix of coefficients, $\xi(t)$ is a time-varying $L \times K$ matrix with unknown continuous dictionary functions entries $\xi_{lk} : \mathcal{T} \to \Re$, and finally $\Sigma_0$ is a positive definite diagonal matrix. Model (1) can be induced by marginalizing out the latent factors $\eta_i$ in

$$y_i = \Theta\xi(t_i)\eta_i + \epsilon_i, \tag{2}$$

with $\eta_i \sim N_K(0, I_K)$ and $\epsilon_i \sim N_p(0, \Sigma_0)$. A generalization includes a nonparametric mean regression by assuming $\eta_i = \psi(t_i) + \nu_i$, where $\nu_i \sim N_K(0, I_K)$ and $\psi(t)$ is a $K \times 1$ matrix with unknown continuous entries $\psi_k : \mathcal{T} \to \Re$ that can be modeled in a related manner to the dictionary elements in $\xi(t)$. The induced mean of $y_i$ conditionally on $t_i = t$, and marginalizing out $\nu_i$ is then

$$\text{E}(y_i|t_i = t) = \mu(t) = \Theta\xi(t)\psi(t). \tag{3}$$

## 1.2 Our modeling contribution

We follow the lead of [22] in using a nonparametric latent factor model as in (2), but induce fundamentally different behavior by carefully modifying the priors $\Pi_\xi$ and $\Pi_\psi$ for the dictionary elements

$\xi_{\mathcal{T}} = \{\xi(t), t \in \mathcal{T}\}$, and $\psi_{\mathcal{T}} = \{\psi(t), t \in \mathcal{T}\}$ respectively. We additionally develop a different and much more computationally efficient approach to computation under this new model.

Fox and Dunson [22] consider the dictionary functions $\xi_{lk}$ and $\psi_k$, for each $l = 1, ..., L$ and $k = 1, ..., K$, as independent Gaussian Processes $GP(0, c)$ with $c$ the squared exponential correlation function having $c(x, x') = \exp(-k||x - x'||_2^2)$. This approach provides a continuous time and flexible model that accommodates missing data and scales to moderately large $p$, but the proposed priors for the dictionary functions assume a stationary dependence structure and hence induce prior distributions $\Pi_\Sigma$ and $\Pi_\mu$ on $\Sigma_{\mathcal{T}}$ and $\mu_{\mathcal{T}}$ through (1) and (3) that tend to under-smooth during periods of stability and over-smooth during periods of sharp changes. Moreover the well known computational problems with usual GP regression are inherited, leading to difficulties in scaling to long series and issues in mixing of MCMC algorithms for posterior computation.

In our work, we address these problems to develop a novel mean-covariance stochastic process with locally-varying smoothness by replacing GP priors for $\xi_{\mathcal{T}} = \{\xi(t), t \in \mathcal{T}\}$, and $\psi_{\mathcal{T}} = \{\psi(t), t \in \mathcal{T}\}$ with nested Gaussian process (nGP) priors [23], with the goal of maintaining simple computation and allowing both covariances and means to vary flexibly over continuous time. The nGP provides a highly flexible prior on the dictionary functions whose smoothness, explicitly modeled by their derivatives via stochastic differential equations, is expected to be centered on a local instantaneous mean function, which represents an higher-level Gaussian Process, that induces adaptivity to locally-varying smoothing.

Restricting our attention on the elements of the prior $\Pi_\xi$ (the same holds for $\Pi_\psi$), the Markovian property implied by the stochastic differential equations allows a simple state space formulation of nGP in which the prior for $\xi_{lk}$ along with its first order derivative $\xi'_{lk}$ and the locally instantaneous mean $A_{lk}(t) = E[\xi'_{lk}(t)|A_{lk}(t)]$ follow the approximated state equation

$$
\begin{bmatrix} \xi_{lk}(t_{i+1}) \\ \xi'_{lk}(t_{i+1}) \\ A_{lk}(t_{i+1}) \end{bmatrix} = \begin{bmatrix} 1 & \delta_i & 0 \\ 0 & 1 & \delta_i \\ 0 & 0 & 1 \end{bmatrix} \begin{bmatrix} \xi_{lk}(t_i) \\ \xi'_{lk}(t_i) \\ A_{lk}(t_i) \end{bmatrix} + \begin{bmatrix} 0 & 0 \\ 1 & 0 \\ 0 & 1 \end{bmatrix} \begin{bmatrix} \omega_{i,\xi_{lk}} \\ \omega_{i,A_{lk}} \end{bmatrix}, \tag{4}
$$

where $[\omega_{i,\xi_{lk}}, \omega_{i,A_{lk}}]^T \sim N_2(0, V_{i,lk})$, with $V_{i,lk} = \text{diag}(\sigma_{\xi_{lk}}^2 \delta_i, \sigma_{A_{lk}}^2 \delta_i)$ and $\delta_i = t_{i+1} - t_i$. This formulation allows continuous time and an irregular grid of observations over $t$ by relating the latent states at $i + 1$ to those at $i$ through the distance $\delta_i$ between $t_{i+1}$ and $t_i$, with $t_i \in \mathcal{T}$ the time observation related to the $i$th observation. Moreover, compared to [23] our approach extends the analysis to the multivariate case and accommodates locally adaptive smoothing not only on the mean but also on the time-varying variance and covariance functions. Finally, the state space formulation allows the implementation of an online updating algorithm and facilitates the definition of a simple Gibbs sampling which reduces the GP computational burden involving matrix inversions from $O(T^3)$ to $O(T)$, with $T$ denoting the length of the time series.

## 1.3 Bayesian inference and online learning

For fixed truncation levels $L^*$ and $K^*$, the algorithm for posterior computation alternates between a simple and efficient simulation smoother step [24] to update the state space formulation of the nGP, and standard Gibbs sampling steps for updating the parametric components of the model. Specifically, considering the observations $(y_i, t_i)$ for $i = 1, ..., T$:

A. Given $\Theta$ and $\{\eta_i\}_{i=1}^T$, a multivariate version of the MCMC algorithm proposed by Zhu and Dunson [23] draws posterior samples from each dictionary element's function $\{\xi_{lk}(t_i)\}_{i=1}^T$, its first order derivative $\{\xi'_{lk}(t_i)\}_{i=1}^T$, the corresponding instantaneous mean $\{A_{lk}(t_i)\}_{i=1}^T$, the variances in the state equations $\sigma_{\xi_{lk}}^2$, $\sigma_{A_{lk}}^2$ (for which inverse Gamma priors are assumed) and the variances of the error terms in the observation equation $\sigma_j^2$ with $j = 1, ..., p$.

B. If the mean process needs not to be estimated, recalling the prior $\eta_i \sim N_{K^*}(0, I_{K^*})$ and model (2), the standard conjugate posterior distribution from which to sample the vector of latent factors for each $i$ given $\Theta$, $\{\sigma_j^{-2}\}_{j=1}^p$, $\{y_i\}_{i=1}^T$ and $\{\xi(t_i)\}_{i=1}^T$ is Gaussian.
Otherwise, if we want to incorporate the mean regression, we implement a block sampling of $\{\psi(t_i)\}_{i=1}^T$ and $\{\nu_i\}_{i=1}^T$ following a similar approach used for drawing samples from the dictionary elements process.

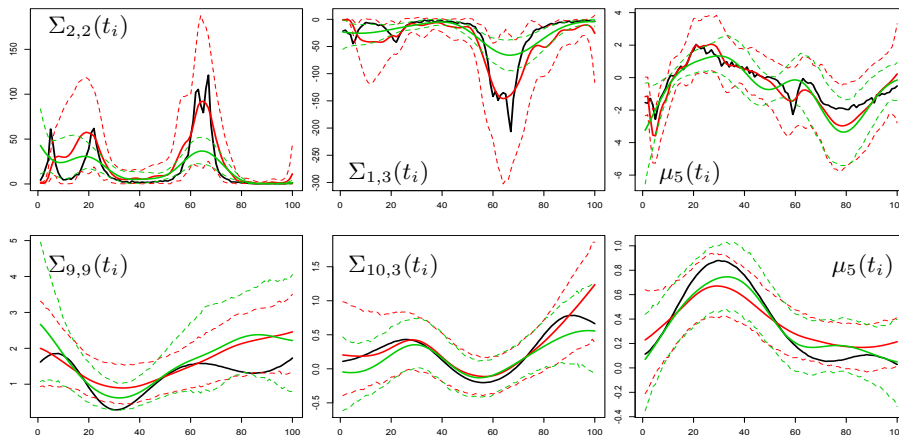

Figure 1: For locally varying smoothness simulation (top) and smooth simulation (bottom), plots of truth (black) and posterior mean respectively of LBCR (solid red line) and BCR (solid green line) for selected components of the variance (left), covariance (middle), mean (right). For both approaches the dotted lines represent the 95% highest posterior density intervals.

C. Finally, conditioned on $\{y_i\}_{i=1}^T$, $\{\eta_i\}_{i=1}^T$, $\{\sigma_j^{-2}\}_{j=1}^p$ and $\{\xi(t_i)\}_{i=1}^T$, and recalling the shrinkage prior for the elements of $\Theta$ defined in [22], we update $\Theta$, each local shrinkage hyperparameter $\phi_{jl}$ and the global shrinkage hyperparameters $\tau_l$ via standard conjugate analysis.

The problem of online updating represents a key point in multivariate time series with high frequency data. Referring to our formulation, we are interested in updating an approximated posterior distribution for $\Sigma(t_{T+h})$ and $\mu(t_{T+h})$ with $h = 1, ..., H$ once a new vector of observations $\{y_i\}_{i=T+1}^{T+H}$ is available, instead of rerunning posterior computation for the whole time series.

Since as $T$ increases the posterior for the time-stationary parameters rapidly becomes concentrated, we fix these parameters at estimates $(\hat{\Theta}, \hat{\Sigma}_0, \hat{\sigma}_{\xi_{lk}}^2, \hat{\sigma}_{A_{lk}}^2, \hat{\sigma}_{\psi_k}^2 \; \hat{\sigma}_{B_k}^2)$ and dynamically update the dictionary functions alternating between steps $A$ and $B$ for the new set of observations. To initialize the algorithm at $T + 1$ we propose to run the online updating for $\{y_i\}_{i=T-k}^{T+H}$, with $k$ small, and choosing a diffuse but proper prior for the initial states at $T-k$. Such approach is suggested to reduce the problem related to the larger conditional variances (see, e.g. [25]) of the latent states at the end of the sample (i.e. at $T$), which may affect the initial distributions in $T + 1$. The online algorithm is also efficient in exploiting the advantages of the state space formulation for the dictionary functions, requiring matrix inversion computations of order depending only on the length of the additional sequence $H$ and on the number of the last observations $k$ used to initialize the algorithm.

## 2 Simulation studies

The aim of the following simulation studies is to compare the performance of our proposal (LBCR, locally adaptive Bayesian covariance regression) with respect to BCR, and to the models for multivariate stochastic volatility most widely used in practice, specifically: EWMA, PC-GARCH, GO-GARCH and DCC-GARCH. In order to assess whether and to what extent LBCR can accommodate, in practice, even sharp changes in the time-varying covariances and means, and to evaluate the costs associated to our flexible approach in settings where the mean and covariance functions do not require locally adaptive estimation tecniques, we will focus on two different sets of simulated data.

The first dataset consists in 5-dimensional observations $y_i$ for each $t_i \in \mathcal{T}_o = \{1, 2, ..., 100\}$, from the latent factor model in (2) with $\Sigma(t)$ defined as in (1). To allow sharp changes of the covariances and means in the generating mechanism, we consider a $2 \times 2$ (i.e. $L = K = 2$) matrix $\{\xi(t_i)\}_{i=1}^{100}$ of time-varying functions adapted from Donoho and Johnstone [26] with locally-varying smoothness (more specifically we choose 'bumps' functions also to mimic possible behavior in practical settings). The second set of simulated data is the same dataset of 10-dimensional observations $y_i$

Table 1: Summaries of the standardized squared errors.

| | **Locally varying smoothness** | | | | **Constant smoothness** | | | |
|---|---|---|---|---|---|---|---|---|
| | mean | $q_{0.9}$ | $q_{0.95}$ | max | mean | $q_{0.9}$ | $q_{0.95}$ | max |
| | covariance $\Sigma(t_i)$ | | | | covariance $\Sigma(t_i)$ | | | |
| EWMA | 1.37 | 2.28 | 5.49 | 85.86 | 0.030 | 0.081 | 0.133 | 1.119 |
| PC-GARCH | 1.75 | 2.49 | 6.48 | 229.50 | 0.018 | 0.048 | 0.076 | 0.652 |
| GO-GARCH | 2.40 | 3.66 | 10.32 | 173.41 | 0.043 | 0.104 | 0.202 | 1.192 |
| DCC-GARCH | 1.75 | 2.21 | 6.95 | 226.47 | 0.022 | 0.057 | 0.110 | 0.466 |
| BCR | 1.80 | 2.25 | 7.32 | 142.26 | 0.009 | 0.019 | 0.039 | 0.311 |
| LBCR | 0.90 | 1.99 | 4.52 | 36.95 | 0.009 | 0.022 | 0.044 | 0.474 |
| | mean $\mu(t_i)$ | | | | mean $\mu(t_i)$ | | | |
| SMOOTH SPLINE | 0.064 | 0.128 | 0.186 | 2.595 | 0.007 | 0.019 | 0.027 | 0.077 |
| BCR | 0.087 | 0.185 | 0.379 | 2.845 | 0.005 | 0.015 | 0.024 | 0.038 |
| LBCR | 0.062 | 0.123 | 0.224 | 2.529 | 0.005 | 0.017 | 0.026 | 0.050 |

investigated in Fox and Dunson [22], with smooth GP dictionary functions for each element of the $5 \times 4$ (i.e. $L = 5, K = 4$) matrices $\{\xi(t_i)\}_{i=1}^{100}$.

Posterior computation, both for LBCR and BCR, is performed by assuming diffuse but proper priors and by using truncation levels $L^* = K^* = 2$ for the first dataset and $L^* = 5, K^* = 4$ for the second (at higher levels settings we found that the shrinkage prior on $\Theta$ results in posterior samples of the elements in the adding columns concentrated around 0). For the first dataset we run 50,000 Gibbs iterations with a burn-in of 20,000 and tinning every 5 samples, while for the second one we followed Fox and Dunson [22] by considering 10,000 Gibbs iterations which proved to be enough to reach convergence, and discarded the first 5,000 as burn-in. In the first set of simulated data, given the substantial independence between samples after thinning the chain, we analyzed mixing by the Gelman-Rubin procedure [27], based on potential scale reduction factors computed for each chain by splitting the sampled quantities in 6 pieces of same length. The analysis shows more problematic mixing for BCR with respect of LBCR. Specifically, in LBCR the 95% of the chains have a potential reduction factor lower than 1.35, with a median equal to 1.11, while in BCR the 95th quantile is 1.44 and the median equals to 1.18. Less problematic is the mixing for the second set of simulated data, with potential scale reduction factors having median equal to 1.05 for both approaches and 95th quantiles equal to 1.15 and 1.31 for LBCR and BCR, respectively.

As regards the other approaches, EWMA has been implemented by choosing the smoothing parameter $\lambda$ that minimizes the mean squared error (MSE) between the estimated covariances and the true values. PC-GARCH algorithm follows the steps provided by Burns [16] with GARCH(1,1) assumed for the conditional volatilities of each single time series and the principal components. GO-GARCH and DCC-GARCH recall the formulations provided by van der Weide [18] and Engle [15] respectively, assuming a GARCH(1,1) for the conditional variances of the processes analyzed, which proves to be a correct choice in many financial applications and also in our setting. Differently from LBCR and BCR, the previous approaches do not model explicitly the mean process $\{\mu(t_i)\}_{i=1}^{100}$ but work directly on the innovations $\{y_i - \hat{\mu}(t_i)\}_{i=1}^{100}$. Therefore in these cases we first model the conditional mean via smoothing spline and in a second step we estimate the models for the innovations. The smoothing parameter for spline estimation has been set to 0.7, which was found to be appropriate to reproduce the true dynamic of $\{\mu(t_i)\}_{i=1}^{100}$. Figure 1 compares, in both simulated samples, true and posterior mean of $\mu(t)$ and $\Sigma(t)$ over the predictor space $\mathcal{T}_o$ together with the point-wise 95% highest posterior density (hpd) intervals for LBCR and BCR. From the upper plots we can clearly note that our approach is able to capture conditional heteroscedasticity as well as mean patterns, also in correspondence of sharp changes in the time-varying true functions. The major differences compared to the true values can be found at the beginning and at the end of the series and are likely to be related to the structure of the simulation smoother which causes a widening of the credibility bands at the very end of the series, for references see Durbin and Koopman [25]. However, even in the most problematic cases, the true values are within the bands of the 95% hpd intervals. Much more problematic is the behavior of the posterior distributions for BCR which badly over-smooth

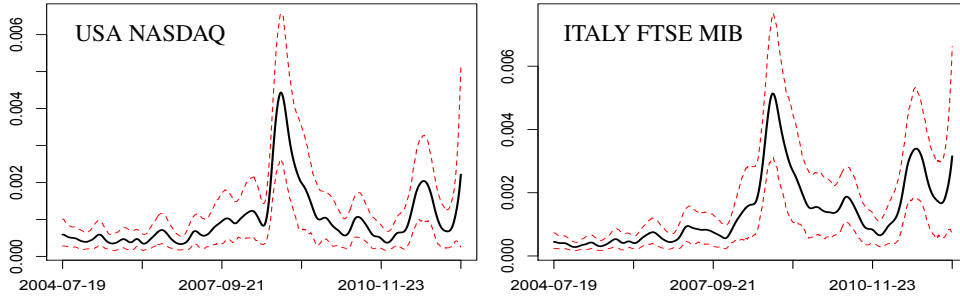

Figure 2: For 2 NSI posterior mean (black) and $95\%$ hpd (dotted red) for the variances $\{\Sigma_{jj}(t_i)\}_{i=1}^{415}$.

both covariance and mean functions leading also to many $95\%$ hpd intervals not containing the true values. Bottom plots in Figure 1 show that the performance of our approach is very close to that of BCR, when data are simulated from a model where the covariances and means evolve smoothly across time and local adaptivity is not required. This happens even if the hyperparameters are set in order to maintain separation between nGP and GP prior, suggesting large support for LBCR.

The comparison of the summaries of the squared errors between true values $\{\mu(t_i)\}_{i=1}^{100}$ and $\{\Sigma(t_i)\}_{i=1}^{100}$ and estimated quantities $\{\hat{\mu}(t_i)\}_{i=1}^{100}$ and $\{\hat{\Sigma}(t_i)\}_{i=1}^{100}$ standardized with the range of the true underlying processes $r_\mu = \max_{i,j}\{\mu_j(t_i)\} - \min_{i,j}\{\mu_j(t_i)\}$ and $r_\Sigma = \max_{i,j,k}\{\Sigma_{j,k}(t_i)\} - \min_{i,j,k}\{\Sigma_{j,k}(t_i)\}$ respectively, once again confirms the overall better performance of our approach with respect to all the considered competitors. Table 1 shows that, when local adaptivity is required, LBCR provides a superior performance having standardized residuals lower than those of the other approaches. EWMA seems to provide quite accurate estimates, however it is important to underline that we choose the optimal smoothing parameter $\lambda$ in order to minimize the MSE between estimated and true parameters, which are clearly not known in practical applications. Different values of $\lambda$ reduces significantly the performace of EWMA, which shows also lack of robustness. The closeness of LBCR and BCR in the constant smoothness dataset confirms the flexibility of LBCR and highlights the better performance of the two approaches with respect to the other competitors also when smooth processes are investigated.

## 3 Application to National Stock Market Indices (NSI)

National Stock Indices represent technical tools that allow, through the synthesis of numerous data on the evolution of the various stocks, to detect underlying trends in the financial market, with reference to a specific basis of currency and time. In this application we focus our attention on the multivariate weekly time series of the main 33 (i.e. $p = 33$) National Stock Indices from $12/07/2004$ to $25/06/2012$ downloaded from `http://finance.yahoo.com`.

We consider the heteroscedastic model for the log returns $y_i \sim N_{33}(\mu(t_i), \Sigma(t_i))$ for $i = 1,...,415$ and $t_i$ in the discrete set $\mathcal{T}_o = \{1,2,...,415\}$, where $\mu(t_i)$ and $\Sigma(t_i)$ are given in (3) and (1), respectively. Posterior computation is performed by using the same settings of the first simulation study and fixing $K^* = 4$ and $L^* = 5$ (which we found to be sufficiently large from the fact that the posterior samples of the last few columns of $\Theta$ assumed values close to 0). Missing values in our dataset do not represent a limitation since the Bayesian approach allows us to update our posterior considering solely the observed data. We run 10,000 Gibbs iterations with a burn-in of 2,500. Examination of trace plots for $\{\Sigma(t_i)\}_{i=1}^{415}$ and $\{\mu(t_i)\}_{t=1}^{415}$ showed no evidence against convergence.

Posterior distributions for the variances in Figure 2 show that we are clearly able to capture the rapid changes in the dynamics of volatilities that occur during the world financial crisis of 2008, in early 2010 with the Greek debt crisis and in the summer of 2011 with the financial speculation in government bonds of European countries together with the rejection of the U.S. budget and the downgrading of the United States rating. Similar conclusions hold for the posterior distributions of the trajectories of the means, with rapid changes detected in correspondence of the world financial crisis in 2008.

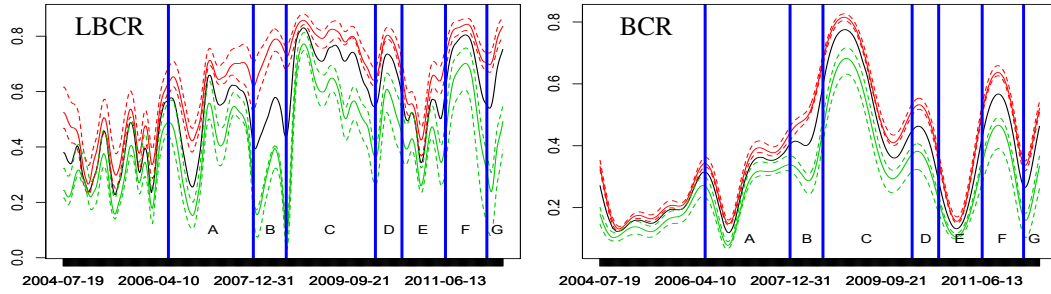

Figure 3: Black line: For USA NASDAQ median of correlations with the other 32 NSI based on posterior mean of $\{\Sigma(t_i)\}_{i=1}^{415}$. Red lines: 25%, 75% (dotted lines) and 50% (solid line) quantiles of correlations between USA NASDAQ and European countries (without considering Greece and Russia). Green lines: 25%, 75% (dotted lines) and 50% (solid line) quantiles of correlations between USA NASDAQ and the countries of Southeast Asia (Asian Tigers and India).

From the correlations between NASDAQ and the other National Stock Indices (based on the posterior mean $\{\hat{\Sigma}(t_i)\}_{i=1}^{415}$ of the covariances function) in Figure 3, we can immediately notice the presence of a clear geo-economic structure in world financial markets (more evident in LBCR than in BCR), where the dependence between the U.S. and European countries is systematically higher than that of South East Asian Nations (Economic Tigers), showing also different reactions to crises. The flexibility of the proposed approach and the possibility of accommodating varying smoothness in the trajectories over time, allow us to obtain a good characterization of the dynamic dependence structure according with the major theories on financial crisis. Left plot in Figure 3 shows how the change of regime in correlations occurs exactly in correspondence to the burst of the U.S. housing bubble (A), in the second half of 2006. Moreover we can immediately notice that the correlations among financial markets increase significantly during the crises, showing a clear international financial contagion effect in agreement with other theories on financial crises. As expected the persistence of high levels of correlation is evident during the global financial crisis between late-2008 and end-2009 (C), at the beginning of which our approach also captures a dramatic change in the correlations between the U.S. and Economic Tigers, which lead to levels close to those of Europe. Further rapid changes are identified in correspondence of Greek crisis (D), the worsening of European sovereign-debt crisis and the rejection of the U.S. budget (F) and the recent crisis of credit institutions in Spain together with the growing financial instability in Eurozone (G). Finally, even in the period of U.S. financial reform launched by Barack Obama and EU efforts to save Greece (E), we can notice two peaks representing respectively Irish debt crisis and Portugal debt crisis. BCR, as expected, tends to over-smooth the dynamic dependence structure during the financial crisis, proving to be not able to model the sharp change in the correlations between USA NASDAQ and Economic Tigers during late-2008, and the two peaks in (E) at the beginning of 2011.

The possibility to quickly update the estimates and the predictions as soon as new data arrive, represents a crucial aspect to obtain quantitative informations about the future scenarios of the crisis in financial markets. To answer this goal, we apply the proposed online updating algorithm to the new set of weekly observations $\{y_i\}_{i=416}^{422}$ from $02/07/2012$ to $13/08/2012$ conditioning on posterior estimates of the Gibbs sampler based on observations $\{y_i\}_{i=1}^{415}$ available up to $25/06/2012$. We initialized the simulation smoother algorithm with the last 8 observations of the previous sample. Plots at the top of Figure 4 show, for 3 selected National Stock Indices, the new observed log returns $\{y_{ji}\}_{i=416}^{422}$ together with the mean and the 2.5% and 97.5% quantiles of their marginal and conditional distributions. We use standard formulas of the multivariate normal distribution based on the posterior mean of the updated $\{\Sigma(t_i)\}_{i=416}^{422}$ and $\{\mu(t_i)\}_{i=416}^{422}$ after 5,000 Gibbs iterations with a burn in of 500. We can clearly notice the good performance of our proposed online updating algorithm in obtaining a characterization for the distribution of new observations. Also note that the multivariate approach together with a flexible model for the mean and covariance, allow for significant improvements when the conditional distribution of an index given the others is analyzed. To obtain further informations about the predictive performance of our LBCR, we can easily use our online updating algorithm to obtain $h$ step-ahead predictions for $\Sigma(t_{T+h|T})$ and $\mu(t_{T+h|T})$ with $h = 1, ..., H$. In particular, referring to Durbin and Koopman [25], we can generate posterior

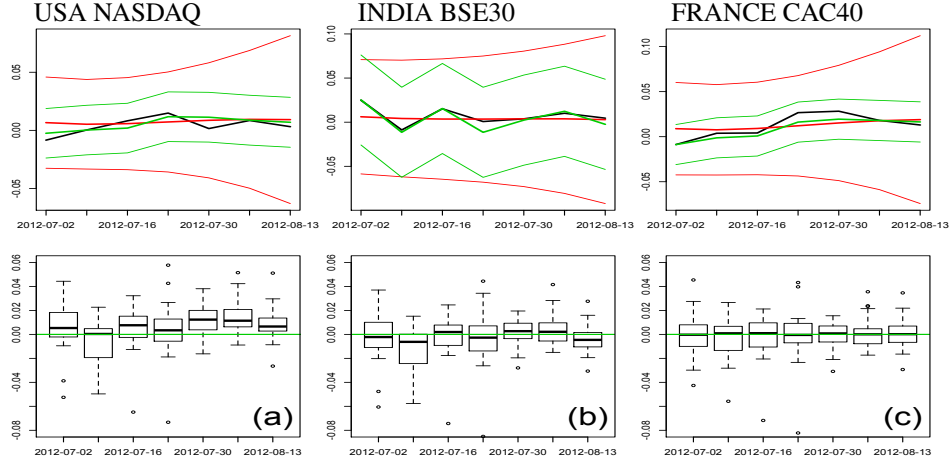

Figure 4: Top: For 3 selected NSI, plot of the observed log returns (black) together with the mean and the 2.5% and 97.5% quantiles of the marginal distribution (red) and conditional distribution given the other 32 NSI (green) $y_{ji}|y_i^{-j}$ with $y_i^{-j} = \{y_{qi}, q \neq j\}$, based on the posterior mean of $\{\Sigma(t_i)\}_{i=416}^{422}$ and $\{\mu(t_i)\}_{i=416}^{422}$ from the online updating procedure for the new observations from $02/07/2012$ to $13/08/2012$. Bottom: boxplots of the one step ahead prediction errors for the 33 NSI computed with 3 different methods.

samples from $\Sigma(t_{T+h|T})$ and $\mu(t_{T+h|T})$ for $h = 1, ..., H$ merely by treating $\{y_i\}_{i=T+1}^{T+H}$ as missing values in the proposed online updating algorithm. Here, we consider the one step ahead prediction (i.e. $H = 1$) problem for the new observations. More specifically, for each $i$ from 415 to 421, we update the mean and covariance functions conditioning on informations up to $t_i$ through the online algorithm and then obtain the predicted posterior distribution for $\Sigma(t_{i+1|i})$ and $\mu(t_{i+1|i})$ by adding to the sample considered for the online updating a last column $y_{i+1}$ of missing values. Plots at the bottom of Figure 4, show the boxplots of the one step ahead prediction errors for the 33 NSI obtained as the difference between the predicted value $\tilde{y}_{j,i+1|i}$ and, once available, the observed log return $y_{j,i+1}$ with $i + 1 = 416, ..., 422$ corresponding to weeks from $02/07/2012$ to $13/08/2012$. In (a) we forecast the future log returns with the unconditional mean $\{\tilde{y}_{i+1}\}_{i=415}^{421} = 0$, which is what is often done in practice under the general assumption of zero mean, stationary log returns. In (b) we consider $\tilde{y}_{i+1|i} = \hat{\mu}(t_{i+1|i})$, the posterior mean of the one step ahead predictive distribution of $\mu(t_{i+1|i})$, obtained from the previous proposed approach after 5,000 Gibbs iterations with a burn in of 500. Finally in (c) we suppose that the log returns of all National Stock Indices except that of country $j$ (i.e. $y_{j,i+1}$) become available at $t_{i+1}$ and, considering $y_{i+1|i} \sim N_p(\hat{\mu}(t_{i+1|i}), \hat{\Sigma}(t_{i+1|i}))$ with $\hat{\mu}(t_{i+1|i})$ and $\hat{\Sigma}(t_{i+1|i})$ posterior means of the one step ahead predictive distribution respectively for $\mu(t_{i+1|i})$ and $\Sigma(t_{i+1|i})$, we forecast $\tilde{y}_{j,i+1}$ with the conditional mean of $y_{j,i+1}$ given the other log returns at time $t_{i+1}$. Prediction with unconditional mean (a) seems to lead to over-predicted values while our approach (b) provides median-unbiased predictions. Moreover, the combination of our approach and the use of conditional distributions of one return given the others (c) further improves forecasts reducing also the variability of the predictive distribution. We additionally obtain well calibrated predictive intervals unlike competing methods.

## 4   Discussion

In this paper, we have presented a generalization of Bayesian nonparametric covariance regression to obtain a better characterization for mean and covariance temporal dynamics. Maintaining simple conjugate posterior updates and tractable computations in moderately large $p$ settings, our model increases the flexibility of previous approaches as shown in the simulation studies. Beside these key advantages, the state space formulation enables development of a fast online updating algorithm useful for high frequency data. The application to the problem of capturing temporal and geo-economic structure between financial markets shows the utility of our approach in the analysis of multivariate financial time series.

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
