[Reviews · NeurIPS 2013]

Submitted by Assigned_Reviewer_3

Multivariate data streams are almost always nonstationary, so methods for tracking and adapting to the local statistics of the timeseries are required. This paper develops and explores a model for local adaptive Bayesian approaches, motivated through definition of a multivariate stoch process. The authors show that such an approach goes some way to avoid computational overload and that it offers performance matching state of the art alternatives.

The paper is well-written, with a clear explanation of the approach. I was able to follow and re-derive the core expressions in the paper, though often this required re-reading other material in the field.

Fig 4 needs more explanation - and some significance testing. "Comparing boxplots in (a) with those in (b) we can see that our model allows to obtain improvements also in terms of prediction." This is very difficult to see and would seem to be stat insig. Please comment or refine the description.
Summary: An interesting paper in an important topic. The theoretical developments are of broad interest, outside the finance domain as well as in it. The choice of results presentation was disappointing after a well-written modeling and development section.

Submitted by Assigned_Reviewer_5

Summary: The paper proposes a multivariate stochastic process for modeling time series which incorporates locally varying smoothness in the mean and in the covariance matrix. The process uses latent dictionary functions with nested Gaussian process priors; the dictionary functions are linearly related to the observations through a sparse mapping. The authors outline MCMC and online algorithms for approximate Bayesian inference and assess performances using simulation and processing of financial data.

Quality: The paper extends the application of the nested Gaussian process priors in [23] to the multivariate case and employs them for both the mean and covariance. This constitutes a sensible extension, and the authors develop an effective inference algorithm.

The authors outline an online algorithm and suggest that it could prove beneficial for high-frequency data. While this is certainly attractive, the paper lacks any clear characterization of the computational overhead (or complexity). Nor is there any indication of the reduction in accuracy that is induced by executing the online method as opposed to rerunning the full posterior computation. As such it is difficult for the reader to gain a sense of the settings in which the online approach might prove useful and appropriate.

The authors do not provide compelling evidence that the extended model proposed in this paper is important in a practical setting. The simulation examples are very toy cases tailored to the technique, so that one would be disappointed if the proposed strategy did not provide improved results. The studies certainly hint at a setting where the local smoothness in the model could prove beneficial, but they do not model a practical measurement setting.

The paper provides an analysis of correlation estimation for national stock indices, with a qualitative analysis of the derived correlation results. Unfortunately, there is limited discussion of the choice of hyperparameters for the inference and as a result it is difficult to determine whether the comparison between LBCR and BCR in Figure 3 is meaningful. The BCR graph has the appearance of a correlation estimate that is constructed using much greater smoothing than the LBCR method. It is not clear that LBCR is using less smoothing where necessary and similar levels of smoothing where appropriate; it just appears to be a much less smooth estimate of the correlation throughout the entire time-series. I wonder whether BCR might achieve a more similar result with different parameter choices.

Since there is no ground truth, it is impossible to know whether the generally increased level of correlation is real or an artifact of the analysis. One cannot know if the oscillations in first portion of the data (2004-2006) are a true reflection of significant variation in the correlations over this period or if they simply indicate that there has been inadequate smoothing in the formation of the estimate. Although the authors provide plausible explanations for the changes in the LBCR estimates, it is not at all clear that these were hypothesized before the results were obtained. If not, then one cannot place much value in the explanations, since it is almost always possible to find some plausible explanation for an increase or decrease in correlation of financial time series at the national level.

The final section of the paper suggests that the model could be employed to obtain better predictions of the log-returns of the national stock indices. The results here add very little of value to the paper. The authors suggest that the boxplots in Figure 4 indicate improved prediction performance, but this figure provides no real evidence of any meaningful improvement. It is extremely unlikely that any analysis conducted on only 7 weeks of log-returns (at the weekly level) would be able to provides any evidence of any improvement in estimation quality.

Clarity: The paper is well-written and easy to follow. At times it is not completely clear what choices have been made in algorithmic simulation and data analysis comparisons.

Originality: The model in the paper is a variation of the model in [22], replacing Gaussian process priors with nested Gaussian process priors. The use of nested Gaussian process priors is suggested in [23], where it is applied to the univariate case, for the mean only. The paper is not significantly original.

Significance: The introduced model is a relatively minor adaptation of an existing model and the inference techniques are fairly straightforward adaptations of existing methods. Although there is some novelty and the model and method are of some interest, the paper is unlikely to have a significant impact.
Summary: The paper is well-written and introduces a novel multivariate model and inference approach, but the innovation is relatively minor. The authors do not provide convincing evidence that the model can provide a meaningful improvement over existing techniques in a practical setting.

Submitted by Assigned_Reviewer_6

The paper proposes a Bayesian method for modeling multivariate (continuous) time series along with a sampling algorithm. The method and algorithm improve existing methods by (1) offering a prior that is locally adaptive to varying smoothness (since existing methods are demonstrated to "under-smooth during periods of stability and over-smooth during periods of sharp changes") and (2) allowing efficient inference due to its formulation using a stochastic differential equation that includes dependence only up to fixed derivative orders (estimates for which, along with the instantaneous mean process A, may then be used as state). The method is examined in both simulation studies, which demonstrate the ability to capture locally varying smoothness compared to other methods, and in an application to stock market indices, which shows both the importance of the modeling regime and the effectiveness of the proposed method.

The paper is well-written and the subject is of significant interest to the NIPS community. However, it seems some aspects of the method could be better explained. The model limitation that makes inference tractable (limiting the derivative dependencies, if I understand correctly, roughly corresponds to the bandwidth truncation used in other methods) could be better highlighted along with the way in which it limits the long-range (and unbounded derivative order) dependencies that general GP modeling allows. Some notation, like the dictionary truncation levels L* and K*, could be more clearly defined, though space constraints are certainly active. (And around lines 106-107, should it read l=1,...,L and k=1,...,K ?) The algorithmic complexity measures could be stated a bit more carefully: getting all means and the full covariance in an unstructured GP may be O(T^3) and require matrix inversion, and that should be pointed out, but many of the competing methods cited (e.g. GPs with truncated bandwidth) scale like O(T). Despite those quibbles, the method seems explained clearly enough to be implemented.

Overall, the paper provides a thorough treatment (up to space constraints) of an interesting new modeling idea that is very relevant to NIPS.
Summary: This paper provides a pretty thorough treatment of an interesting new time series modeling method that is of significant interest to the NIPS community.
Author Feedback

Author rebuttal: Novelty & Significance:

Our proposed approach is novel and significant in being the first coupled mean-covariance process, which allows locally varying smoothness. Although our work builds on the (unpublished) Fox and Dunson (2011) formulation, their approach assumes a single level of smoothness over time, and hence is substantially less flexible than ours. We accomplish this flexibility by using nested Gaussian processes, which have only been considered previously by Zhu and Dunson (2012) in an unpublished arXiV manuscript focused on single function estimation. We additionally develop efficient computational algorithms, reducing the O(T^3) bottleneck of GPs to O(T) and obtaining an accurate online algorithm
as an alternative to MCMC. Our methodology can be broadly used and improve on the state of the art in multivariate time series settings and beyond.



Technical Comments

1] Online: Our algorithm is not fully online in updating on the time varying dictionary functions. As T increases the posterior for the time-stationary parameters rapidly becomes concentrated, so it is reasonable to fix these parameters at estimates while dynamically updating the dictionary functions. We validate this algorithm in simulation studies, showing that our online approximation to the posterior yields accurate results and predictions. The online algorithm is also efficient in exploiting the advantages of the state-space formulation for the dictionary functions. We need matrix inversion computations of order depending only on the length of the additional sequence H and on the number of the last observations k used to initialize the algorithm O((T+H)-(T-k)) = O(H+k), a massive reduction.

2] Simulation Study: The simulated datasets are not tailored for our model since the dictionary functions are time-varying functions adapted from Donoho and Johnstone (1994), instead of being generated from nGP. Moreover the structure of the underlying mean and covariance processes has been chosen also to mimic possible behavior in practical settings. The “bumps” in the covariance functions from the simulated dataset are also found in the estimated volatility processes in Figure 2 using real data. Similar bumps are observed in protein mass spectrometry, influenza levels at different locations over time, and electricity load trends (see e.g. Ba et al., 2012).

3] Application: There is a rich theory literature supporting usual GPs having careful hyperpriors on the covariance parameters, including posterior consistency (Ghosal and Roy, 2006) and minimax optimal rates of posterior concentration (van der Vaart and van Zanten, 2008). However, this theory assumes that the function is in a smooth class, with a single smoothness level at all locations. If the smoothness varies locally, GPs with a stationary covariance will yield badly sub-optimal rates. Our simulations for the Fox and Dunson (2011) approach illustrate how this sub-optimality can lead to poor
performance in applications.
We observe that the posteriors for parameters characterizing the nGP dictionary functions concentrate on values consistent with varying smoothness, even when priors for these parameters are noninformative. We learn important new aspects of the data, which were not previously apparent and do not show up with alternative analysis methods. In particular, the change of regime as well as the most evident increase in correlation occur in correspondence with financial events of main importance worldwide rather than on national events. This is consistent with the “international contagion effect” theory of financial markets (Baig and Goldfajn, 1999 and Claessens and Forbes, 2009) and also with recent applications of stochastic volatility models to exchange rates (see e.g. Kastner et al. 2013).

4] Prediction: prediction with unconditional mean in a] seems to lead to over-predicted values while our approach b] seems at least to provide median-unbiased predictions. The combination of our approach and the use of conditional distributions of one return given the others c] further improves predictions reducing also the variability of the predictive distribution. We additionally obtain well calibrated predictive intervals unlike competing methods.



REFERENCES

- Ba, A., Goude Y., Sinn M., & Pompey P. (2012). “Adaptive Learning of Smoothing Functions: Application to Electricity Load Forecasting.” In NIPS (Neural Information Processing Systems).

- Baig, T., & Goldfajn, I. (1999). “Financial Market Contagion in the Asian Crisis.” Staff Papers, International Monetary Fund, 46, 167-195.

- Claessens, S., & Forbes, K. (2009). International Financial Contagion, An overview of the Issues. Springer.

- Donoho, D.L., & Johnstone, J.M. (1994). “Ideal spatial adaptation by wavelet shrinkage.” Biometrika, 81, 425-455.

- Fox, E., & Dunson, D.B. (2011). “Bayesian Nonparametric Covariance Regression.” arXiv:1101.2017.

- Ghosal, S., & Roy, A. (2006). “Posterior consistency of Gaussian process prior for nonparametric binary regression.” The Annals of Statistics,34, 2413-2429.

- Kastner, G., Früwirth-Schatter, S., & Lopes H.F., (2013). “Efficient Bayesian inference for multivariate factor stochastic volatility models”. In BAYSM2013 (Bayesian Young Statistician Meeting).

- Van der Vaart, A.W., & Van Zanten, J.H. (2008). “Rates of contraction of posterior distributions based on Gaussian process priors.” The Annals of Statistics, 36, 1435-1463.

- Zhu, B., & Dunson, D.B. (2012). “Locally Adaptive Bayes Nonparametric Regression via Nested Gaussian Processes.” arXiv:1201.4403.